# Drag Coefficient of Submerged Flexible Vegetation Patches in Gravel Bed Rivers



**Kourosh Nosrati [1], Hossein Afzalimehr [1,\*] and Jueyi Sui [2]**

[1] Natural Disasters Prevention Research Center, School of Civil Engineering, Iran University of Science and Technology, Tehran 16846-13114, Iran; koursoh_nosrati@civileng.iust.ac.ir

[2] School of Engineering, University of Northern British Columbia, Prince George, BC V2N 4Z9, Canada; jueyi.sui@unbc.ca

[\*] Correspondence: hafzali@iust.ac.ir; Tel.: +98-913-2175524

**Abstract:** Vegetation patches and strips either along riverbanks or in channel beds are essential for the protection of erosion and sedimentation processes. In the present study, the drag coefficient $C_{dv}$ of submerged flexible vegetation patches in gravel bed rivers was investigated. A total of 13 vegetation patches with different densities were studied in disparate reaches of the Padena Marbor and Beheshtabad gravel bed rivers in Iran. Water depths, flow velocities, and particle grain sizes around these vegetation patches were collected. The Saint-Venant equation and various empirical equations for estimating the drag coefficient were applied to study hydrodynamics in the presence of vegetation patches under nonuniform flow conditions. Furthermore, the drag coefficient factor of flexible vegetation was used to represent the flexibility of vegetation patches and drag characteristics, which were explored from the perspective of material mechanics. The results showed that the calculated values of $C_{dv}$ exhibited nonuniform variations with the increase in the Reynolds number along the streamwise direction due to flow nonuniformity. Two effects caused by flexible vegetation patches were observed, namely, the sheltering effect (for $Re_d > 580$) and blockage effect (for $Re_d < 450$). In most of the vegetated patches, the sheltering effect was dominant, which reduced the drag coefficient. Finally, a fitting formula was proposed based on the drag coefficient factor and Cauchy number.

**Keywords:** gravel bed river; drag coefficient; submerged flexible vegetation patches; Saint Venant; Cauchy number

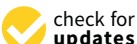



## 1. Introduction

Aquatic vegetation in riverbeds and riverbanks has a crucial effect in aquatic ecosystems. For instance, plants play an important role in transporting contaminants through changes in flow hydrodynamics. The interaction between flow and vegetation should be considered in the projects of urban hydrology, landscape architecture infrastructures, stream restoration, and flood management. Enrichment and development of vegetation patches have numerous benefits for the environment [1], indicating that plants have a remarkable role in erosion control in addition to their ecological effects compared to structural methods. To better understand the interaction between flow and vegetation patches, the concept of drag forces due to rigid and flexible elements should be studied. Using different parameters to estimate the drag coefficient of vegetation [2], researchers have claimed that there is a significant difference in velocity and Reynolds stress distributions for emergent and submerged vegetation cover [3–8]. Some studies have focused on specific aspects of vegetation such as foliage impacts [9], effects of stem flexibility on turbulence [10], and the mechanical behavior of vegetation [11]. Certain models have been proposed to take into account the effect of vegetation on the flow resistance [12] and total hydraulic resistance [13]. These models are mostly related to specific vegetation morphologies and are not easily applied to other vegetation species or morphologies. Cheng (2011) proposed

a representative roughness height to describe the resistance of vegetated open channel flows [11]. It was reported that the friction factor determined for the surface layer above the vegetation normally increases with the relative roughness ($d_{50}/H$). Tang et al. (2014) studied drag coefficients in the presence of submerged stems in an open channel flow and proposed an empirical formula for three drag coefficients (the drag coefficient for an isolated cylinder, the bulk drag coefficient of an array of cylinders, and the local drag coefficient) [14]. The developed semi-analytical relationship of flow resistance indicates that the depth of validity logarithmic law depends on the location of measurements, including both the pool inlet and outlet in the presence of vegetation. The results indicate that, in the presence of vegetation in channel beds, the validated depth of log law affects the flow resistance because the drag coefficient is influenced by shear stress estimation due to vegetation. Although these studies used different vegetation patches in size and geometry as either emergent or submerged in their setup, the results of these studies show that the presence of vegetation causes deviations in the main hydraulic parameters including velocity and Reynolds stress distributions from the classic ones under uniform flow conditions. In addition, the presence of vegetation in the channel bed or channel bank affects the location of the maximum velocity.

In the last decade, vegetation-induced drag has attracted much attention from many researchers and river engineers by means of laboratory experiments, numerical simulations, and field observations. To quantify the vegetation-induced resistance, the drag coefficient is normally examined by assuming that energy losses occur via the distributed drag forces of vegetation elements, and it is not just limited to the vegetated drag using the bed stress formulae based on previous studies. The drag coefficient $C_d$, which is a function of the time-averaged drag force $F_d$ on vegetation elements, the water density $\rho$, the element width $D$, the flow velocity $U$, and the element height $h$ in the vertical direction, is defined as [15]:

$$C_d = \frac{F_d}{\rho U^2 A_p / 2} \tag{1}$$

where $A_p$ is the frontal projected area. Some researchers have claimed that $C_d$ is a function of the vegetation density [16], the stem Reynolds number [14], and the vegetation flexibility [17]. Up until now, to estimate $C_d$, the vegetation elements have normally been considered to be rigid cylinders with a steady uniform flow regime. Cheng (2013) proposed a relationship between the drag coefficients and Reynolds number based on the concept of the pseudo-fluid, wherein $C_d$ generally decreases with the increase in Reynolds number from 1 to $1 \times 10^4$, which applies to the rigid emergent elements in steady uniform flows [18]. Clearly, a vegetation patch cannot be simplified to consist of numerous rigid cylindrical elements, as the aquatic vegetation has various flexibility and morphological complexities. Consequently, it is difficult to estimate the drag coefficient considering the flexibility of vegetation in the riverbed. As both $A_p$ and $C_d$ are functions of flow velocity [19], an applied model will be needed to estimate the drag coefficient by considering the flexibility of vegetation with high accuracy. Unlike rigid elements, the degree of bending of flexible vegetation elements depends on the flow conditions [20–22]. In addition, the bending deformation of flexible vegetation leads to changes in $A_p$ and the shape of the momentum absorption, which ultimately causes the momentum to transfer to the channel bed and reduce the current resistance [23]. Vogel (1989) used the term of reconfiguration to describe the deformation of flexible plants [24]. This reconfiguration reduces the pressure drag and thus affects the flow structure in the flexible vegetation. For the flexible vegetation, the spatially averaged drag force is defined as [24]:

$$F_{d\chi} = \frac{1}{2}\rho A_f \left(\frac{1}{U_b^\chi}\right) C_{Db} A_p U^{(2+\chi)} \tag{2}$$

where $F_{d\chi}$ is the drag force of flexible vegetation, $A_f$ is the number of plants per unit bed area, $\chi$ is a parameter that accounts for the reconfiguration of flexible plants, $U_b$

is a reference velocity, and $C_{Db}$ is the form drag coefficient. Equation (2) is a quantitative description of the reconfiguration and drag coefficient reduction for a vegetated stream [22]. The mechanical behavior of flexible vegetation has been proposed to estimate the vegetation-induced drag force [25,26] and total flow dynamic resistance [27]. Whittaker et al. (2015) presented the Cauchy number $C_s$ to determine the drag coefficient and reconfiguration of flexible vegetation [28]. The effect of flexible vegetation-induced reconfiguration $(U^\chi / U_b^\chi) C_{Db}$ is described by means of $C_s$ through a function of the modulus of elasticity $E_V$. The abovementioned studies were conducted under conditions of uniform flows, while the flow is completely nonuniform in natural channels having vegetated patches. Generally, for the flow over gravel-bed streams with flexible vegetation patches, the flow is assumed to be steady and quasi-uniform, implying a quasi-equilibrium condition of the forces between the flow-driving term $\left[ \gamma B H d_x (1 - \varnothing_{veg}) S_f \right]$ and the resistance term $[B F_d d_x] + [B d_x (1 - \varnothing_{veg}) \tau_{bed}] + [2 H d_x \tau_{wall}]$ for a given small length scale $d_x$, where $B$ is the channel width, $\varnothing_{veg} = \pi D^2 / \Delta s^2$ is vegetation density [2], $\Delta s$ is the average spacing distance between the adjacent stems, $D$ is the diameter of individual vegetation stems, $\tau_{bed}$ is the bed shear stress, $\tau_{wall}$ is the wall shear stress, $H$ is the flow depth, $\gamma$ is the water-specific gravity, and $S_f$ is defined as the total energy head loss per unit stream. In fact, averaging the values of measurements at each local point leads to the steady flow case, and if this condition remains stable along the study channel reach, the flow uniformity will be satisfied as well. However, we have

$$\gamma B H (1 - \varnothing_{veg}) S_f d_x = [B F_d d_x] + \left[ B (1 - \varnothing_{veg}) \tau_{bed} d_x \right] + [2 H \tau_{wall} d_x] \tag{3}$$

For dense vegetated patches, as reported by [2], the drag force caused by vegetated patches on the right-hand side of above equation is much larger than the friction caused by the channel bed and sidewall.

$$\gamma H (1 - \varnothing_{veg}) S_f = F_d \tag{4}$$

The vegetation-induced resistance increases the residence time of the flow within the vegetated zone and finally increases the cumulative infiltration [29]. The nonuniform behavior of flexible vegetation occurs in the channel with a high vegetation density. In rivers, the presence of various three-dimensional bedforms and bed materials sustains the effect of nonuniform flow [30]. Wang et al. (2015) investigated the nonuniform flow within rigid emergent cylindrical vegetation, and proposed an empirical equation to estimate the flow resistance based on the Saint-Venant equation [31]. Whenever the flow current passes the flexible vegetated patches, the flexible vegetated elements show a bending deformation, resulting in drag reconfiguration [32]. This phenomenon often happens in natural rivers with flexible vegetation patches. The flow resistance is more complex than that of rigid cylindrical elements used in a laboratory.

In the presence of flexible submerged vegetation, the most important characteristic of flow is the development of a shear-layer at the top of the vegetation [33,34]. The shear-layer generates coherent vortices by the Kelvin—Helmholtz (KH) instability. Early studies of sediment transport in a flow with the presence of vegetation commonly focused on the role of vegetation resistance on bed shear stress. In fact, the total shear stress at the top of the vegetation could be considered as the sum of a bed shear stress and a vegetation drag force. Actually, the additional drag induced from vegetation results in a decrease in the mean flow and the bed shear stress within vegetated regions compared to those without vegetation in an open channel [35]. Houser et al. (2015) found that the relation between the drag coefficient and Reynolds number depends on the flexibility and morphology of vegetation, and more flexibility causes more reduction in the drag coefficient [33]. In addition, they reported that flexible vegetation has a small drag coefficient compared to rigid vegetation, and the prediction of the drag coefficient by different models depends on the methodology and vegetation morphology [33]. Luhar et al. (2017) showed that vegetation flexibility results in a much lower drag compared to rigid blades. They also found that wave decay

increases when the vegetation occupies a larger part of flow depth [34]. The results of experiments by Carevallaro et al. (2018) showed a complex pattern between the drag coefficient and Reynolds number due to flexible leaves. They found that the mean flow velocity inside the vegetation is lower than that estimated above the vegetation cover. This is due to the interaction of vegetation and oscillatory velocity [35].

Considering the above studies, one can say that flexible vegetation plays a significant role in the prediction of the drag coefficient. Accordingly, the more flexible the vegetation, the greater the reduction in drag coefficient. In addition, most studies, especially those published in recent years, have focused on the interaction of waves and flexible vegetation, which is very important in coastal engineering. However, it is necessary to investigate the interaction of flow nonuniformity rather than wave characteristics in river engineering projects. In fact, the flow conditions in rivers are different from those in coastal engineering, emphasizing more research in this area.

Morphologically, flexible plants are significantly different from rigid elements, used in most laboratory works. The main purpose of this study is to predict the effect of reconfiguration of the flexible vegetation patch under submerged conditions by means of the material mechanics and Cauchy number in gravel bed rivers. Specifically, the interactions between the flexibility of vegetation and flow nonuniformity are investigated. Based on the Saint-Venant equation, a formula is proposed to estimate the drag coefficient for submerged flexible vegetation patches. In addition, the difference between the rigid and flexible vegetation covers with the purpose of predicting the effect of reconfiguration is discussed. A relationship based on the Cauchy number is developed. All the analyses presented in this study are limited to submerged flexible vegetation patches in gravel bed rivers.

## 2. Theory

### 2.1. Saint-Venant Equation

For a steady nonuniform flow with flexible vegetated patches in open channels, it is normally assumed that the flow is locally steady and uniform, achieving an equilibrium between the forces of the current-driving term and the resistance term. The motion of flow through submerged vegetation patches is normally treated as one-dimensional. Under such flow conditions, the Saint-Venant equation for an open channel flow through vegetation is given by [31]

$$\frac{\partial U}{\partial t} + U\frac{\partial U}{\partial x} + g\left(\frac{\partial H}{\partial t} + S_f - S_0\right) = 0 \tag{5}$$

where $t$ is the time and $x$ is the streamwise direction of the flow, $S_0$ is the bed slope, and $g$ is the acceleration of gravity. The other parameters have been defined before.

### 2.2. Flexible Vegetated Drag Coefficient under Submerged Conditions

Substituting Equation (1) into Equation (4) with $A_p = DH(x)/\Delta s^2$:

$$S_f = \left[\frac{C_{dv}D}{(1 - \varnothing_{veg})\Delta s^2}\right]\frac{U(x)^2}{2g} \tag{6}$$

where $C_{dv}$ is the drag coefficient for flexible submerged vegetated patches and $U(x)$ is the flow velocity of the streamwise direction obtained from:

$$U(x) = \frac{Q}{B(1 - \varnothing_{veg})H(x)} \tag{7}$$

Substituting Equations (6) and (7) into Equation (5) obtains the drag coefficient formula for the submerged flexible vegetal elements and yields

$$C_{dv} = 2g(1 - \varnothing_{veg})\frac{\Delta s^2}{D}\left[\frac{S_0}{U^2} + (\frac{1}{U^2} - \frac{\Delta s^2}{gH(x)})(-\frac{\partial H}{\partial x})\right] \tag{8}$$

In this equation, $E_{veg} = 2g(1 - \varnothing_{veg})\Delta s^2/D$ is affected by the vegetation properties such as the density, diameter, and the average spacing distance between vegetation elements. However, this equation is independent of the flow components [36]. These researchers stated that $H(x)$ is the key parameter for describing the flow characteristics in Equation (8) that varies in the streamwise direction of the flow $x$. $H(x)$ can be expressed as a quadratic function within emergent vegetation [31].

$$H(x) = ax^2 + bx + c \;\rightarrow\; \frac{\partial H}{\partial x} = 2ax + b \tag{9}$$

where $\partial H/\partial x$ is defined as the derivative of $H$ to $x$ and a regression function is used to determine $b$ and $c$. However, as shown in the following, $H(x)$ also changes for flexible vegetated elements. There are limitations for using Equation (9): For experiments with large hydraulic gradients, Equation (9) can be used to estimate $C_{dv}$ in Equation (8). It is found that by reducing the hydraulic gradient and the nonuniformity of the flow, the errors of using this equation increase. This means that the application of the quadratic regression for laboratory experiments with low hydraulic gradients will not be appropriate [31].

### 2.3. Cauchy Number $C_s$

In addition to the effects of drag, flexible vegetation is subject to restoring forces due to the vegetation rigidity [37]. In this study, to measure $D$ and $\Delta s$ in Equation (8), all vegetation elements in each patch were pruned and the height of each element was identified. The focus of this study is mainly on the restoring force acting on the drag reconfiguration of flexible vegetation. By neglecting the effect of buoyancy, the bending deformation of flexible vegetation can be considered as a rigid cylindrical element. In the reported studies, the drag coefficient factor, defined as $\alpha_b = C_{dv}/C_D$, was considered as the effect of bending deformation on flexible vegetation compared to that on rigid vegetation [38]. The modulus of elasticity of each vegetation element $E_s$ is known as the Young's modulus, which describes its bending deformation. Another parameter is the Cauchy number, which is used to determine the degree of flexibility of a rigid cylindrical element. This parameter is defined as the ratio of the dynamic pressure to the Young's modulus ($E_s$, N/m$^2$) [39],

$$C_s = \frac{\rho U^2}{E_s} \tag{10}$$

To estimate the Young's modulus in Equation (10), the following nonlinear equation with 70% accuracy in rivers is used.

$$E_s = 7.648 \times 10^6 (\frac{\overline{h}_p}{D}) + 2.174 \times 10^4 \left(\frac{\overline{h}_p}{D}\right)^2 + 1.809 \times 10^3 (\frac{\overline{h}_p}{D})^3 \tag{11}$$

## 3. Material and Methods
### 3.1. Study Site

All field measurements of this study were conducted in the Beheshtabad (Figure 1a) and Marbor (Figure 1b) rivers in Iran. Both rivers are some of the important tributaries of the Karun River in Iran. These reaches are located in Chaharmahal Bakhtiari and Esfahan Provinces in Iran, respectively.

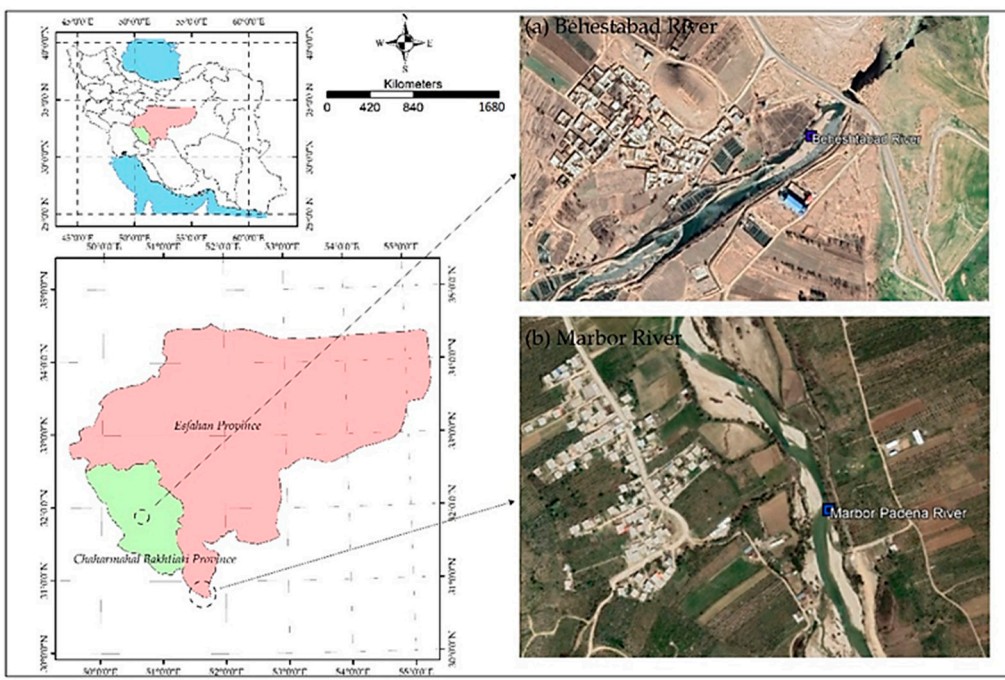

**Figure 1.** Location of study sites (**a**) Beheshtabad River and (**b**) Marbor River.

*3.2. Characteristics of Vegetated Patches*

The submerged flexible vegetation patches in these reaches were the Charophyta algae (a group of freshwater green algae), as shown in Figure 2a,b in both rivers.

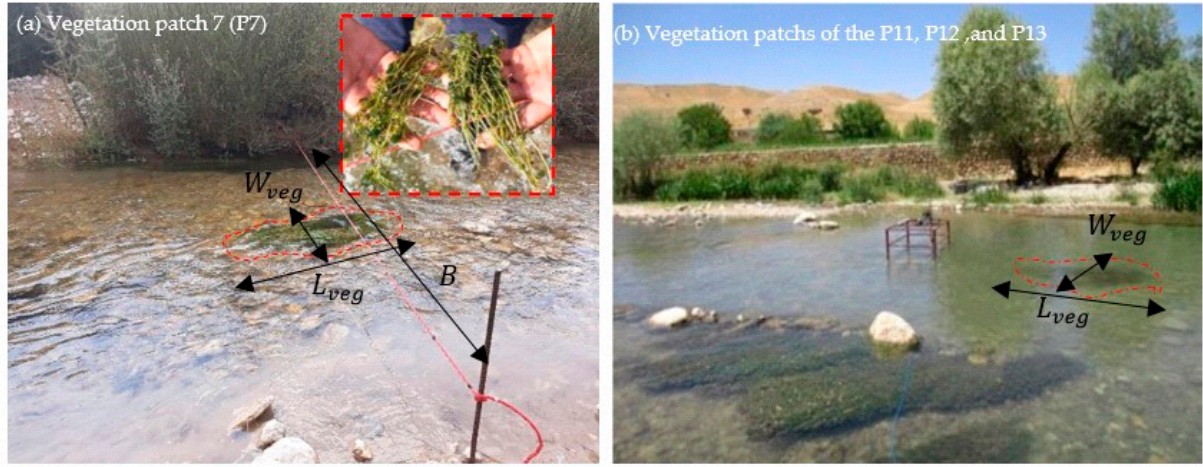

**Figure 2.** (**a**) Vegetation patch (P7) in one of the selected reaches of the Padena Marbor River. (**b**) Vegetation patches of P11, P12, and P13 in Beheshtabad River.

In Figure 2a, the maximum length (streamwise) and width (spanwise) of the vegetated patch are expressed as $L_{veg}$ and $W_{veg}$, respectively, and $B$ is the river width. It should be noted that the accurate measurement of the dimensions of the geometric properties of each vegetated patch, including $D$, $\Delta s$, and $\overline{h}_p$, is very difficult and complex in the rivers. Thus, in this study, each vegetated patch was pruned to convert each vegetated element into cylindrical form, showing the proposed average values assuming that a rigid cylindrical element is effective in estimating the drag coefficient, as it was considered in the study of Jahadi et al. [40]. The average values were used to present the vegetation cover characteristics because it is practically impossible to measure them for each stem of vegetation. The dimensions of vegetal elements of each patch were measured using a

caliper with an accuracy of 0.1 mm. All geometric properties of vegetation patches are shown in Table 1. In the Beheshtabad River, three vegetation patches were identified in this study, namely, P11, P12, and P13. The other vegetation patches were in the Padena Marbor River, in Table 1. The minimum and maximum density of each vegetated patch were estimated with values of $14.2 \times 10^{-3}$ to $65.5 \times 10^{-3}$, respectively.

**Table 1.** Geometric properties of vegetation patches.

| Patch | $L_{veg}$ (m) | $W_{veg}$ (m) | $\bar{h}_p$ (m) | $D$ (mm) | $\Delta s$ (mm) |
|---|---|---|---|---|---|
| P1 | 1.05 | 0.013 | 0.048 | 1 | 4.8 |
| P2 | 0.59 | 0.27 | 0.055 | 0.8 | 5.2 |
| P3 | 0.24 | 0.09 | 0.047 | 1.1 | 5.6 |
| P4 | 1.46 | 0.47 | 0.066 | 0.8 | 5 |
| P5 | 0.67 | 0.29 | 0.1 | 1.3 | 4.5 |
| P6 | 1.07 | 0.53 | 0.16 | 1.1 | 4.5 |
| P7 | 1.08 | 0.72 | 0.21 | 0.82 | 5.1 |
| P8 | 4 | 0.25 | 0.12 | 0.9 | 5 |
| P9 | 0.4 | 0.15 | 0.05 | 0.8 | 5.2 |
| P10 | 1.2 | 0.38 | 0.03 | 0.7 | 5.2 |
| P11 | 0.96 | 0.72 | 0.28 | 0.9 | 4.45 |
| P12 | 0.96 | 0.72 | 0.085 | 0.9 | 5.22 |
| P13 | 0.96 | 0.72 | 0.08 | 0.9 | 5.77 |

The main variable measured here was the water surface profile $H(x)$ along each vegetated patch from the upstream to downstream of each patch. For each vegetated patch, $x = 0$ denotes the starting point of the flow into the vegetated zone (and normalized as $x^+ = x/L_{veg}$). The normalized $H^+(x^+) = [H(x) - H_0]/(H_i - H_0)$ was employed to characterize the nonuniform water surface profile, where $H_i = H(0)$ was measured at 10 cm upstream of the vegetated patch edge and $H_o = H(L_{veg})$ is the smallest water depth at 10 cm downstream of the vegetated patch. The values of $\varnothing_{veg}$, $E_{veg}$, $H_i$, and $H_o$, as well as the fitted parameters a, b, and c, are summarized in Table A1 in Appendix A. It should be noted that $H_1$ to $H_5$ are the measured water depths from $x = 0$ (the upstream edge of vegetation patch) to $x = L_{veg}$ (the downstream edge of vegetation patch). The flow rate $Q$ is variable for each vegetated pach. The variable flow rate for each vegetated pach is due to the upstream water usage such as irrigation. This phenomenon was observed at vegetation patches of P1 to P7 in the Padena Marbor River.

### 3.3. Experimental Setup and Duration of the Data Acquisition

All field data were collected in Beheshtabad and Marbor Padena rivers from June to August 2020. The important hydraulic parameters and data collected in these rivers are summarized Table 2. where $L_R$ is the length of river reaches, $Q$ is the flow rate, $Fr = U/(gH)^{0.5}$ is the flow Froude number, and $Re = UH/\vartheta_m$ is the Reynolds number. In this study, the kinematic viscosity of water ($\vartheta_m$) at 25 °C was $0.895 \times 10^{-6}$ m$^2$/s.

**Table 2.** Hydraulic parameters.

| River | $L_R$ (m) | $H$ (m) | $U$ (m/s) | $B$ (m) | $Q$ (m$^3$/s) | $Fr$ | $Re$ ($\times 10^3$) |
|---|---|---|---|---|---|---|---|
| Beheshtabad | 3.5 | 0.16 | 0.3 | 7 | 0.99 | 0.2 | 116 |
| Padena Marbor | 14.25 | 0.15–0.21 | 0.8–1.0 | 3.37–5.2 | 0.54–0.87 | 0.59–0.85 | 150–182 |
| | 10 | 0.26–0.64 | 1.0–1.1 | 1.8–14.6 | 1.8–3.09 | 0.44–0.5 | 424–698 |
| | 8.5 | 0.16–0.4 | 0.9–1.3 | 2.03–9.7 | 0.85–2.18 | 0.62–1.09 | 218–428 |

The flow velocity was measured by using a Butterfly Current Meter (BCM) with an accuracy of 0.1 m/s. The time of velocity recordings at each point was 50 s, and the measurements at each point were repeated three times. The BCM was constructed based

on the relationship between water flow velocity ($U$) and the axial rotation speed ($N$), i.e., $U = a + bN$, where $a$ and $b$ are fixed coefficients determined by its manufacturer. The BCM is equipped with a counter with an accuracy of a second and can show $N$ per second. According to the value of $N$, the values of $a$ and $b$ are extracted from a specific table and then the value of $U$ is calculated. The flow rate $Q$ was calculated by using the continuity equation $Q = \sum_{i=1}^{n} u_i A_i$ where $A_i$ is the cross-sectional area and $u_i$ is the mean velocity in each sub-cross-section. In addition, the flow depth was measured with a ruler with an accuracy of 1 mm. Figure 3 presents the grain size distribution of bed material around each vegetation patch (P1 up to P13) in the Padena Marbor and Beheshtabad rivers, obtained by using the traditional technique [41].

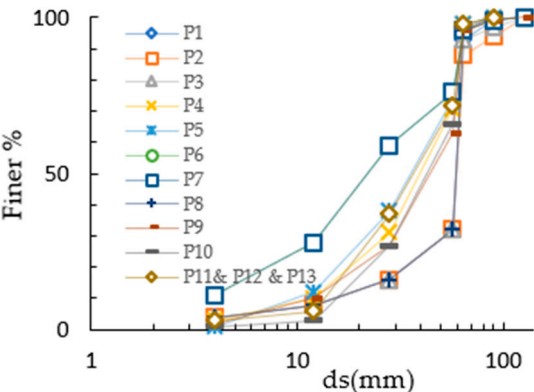

**Figure 3.** Grain size distribution of bed material around vegetation patches.

The procedure can be described as follows: (a) collection of data at the selected section of river reach; (b) measurement of the particle size and assessment of grain size distribution by using Wolman method [41] across the selected section (Figure 3). For example, the median grain size of bed material around vegetation patch 7 (P7) and geometric standard deviation ($\sigma_g = (d_{84}/d_{16})^{0.5}$) was 18 mm and 2.26, respectively, according to Figure 1, where $d_{84}$ and $d_{16}$ are the 16th and 84th percentile of a particle-size distribution, respectively. For all vegetation patches, $\sigma_g$ was more than 1.4. This means that the grain size distribution around all vegetation patches was completely no-monotonous [8]. The grain size distribution at P11, P12, and P13 was nearly the same and the only difference between them was their density, which changed after being pruned (Figure 2b).

## 4. Results

### 4.1. Water Surface Profile

In this study, the measured water surface profile of nonuniformity was modeled by the quadratic function (i.e., Equation (9)), and the appropriate fitting parameters are summarized in Table A1. Figures 4 and 5 show a clear agreement ($R^2 > 0.9$) between the measured (solid points) and fitted (dashed line) values $H(x)$ over all vegetation patches with various densities. In these figures, the abscissa and the ordinate describe the normalized distance by $L_{veg}$ and the flow depth with $H^+(x^+) = [H(x) - H_0]/(H_i - H_0)$, respectively. As shown in Figure 4, for all the patches, the value of $H(x)$ from the upstream edge to the downstream edge of each patch decreases.

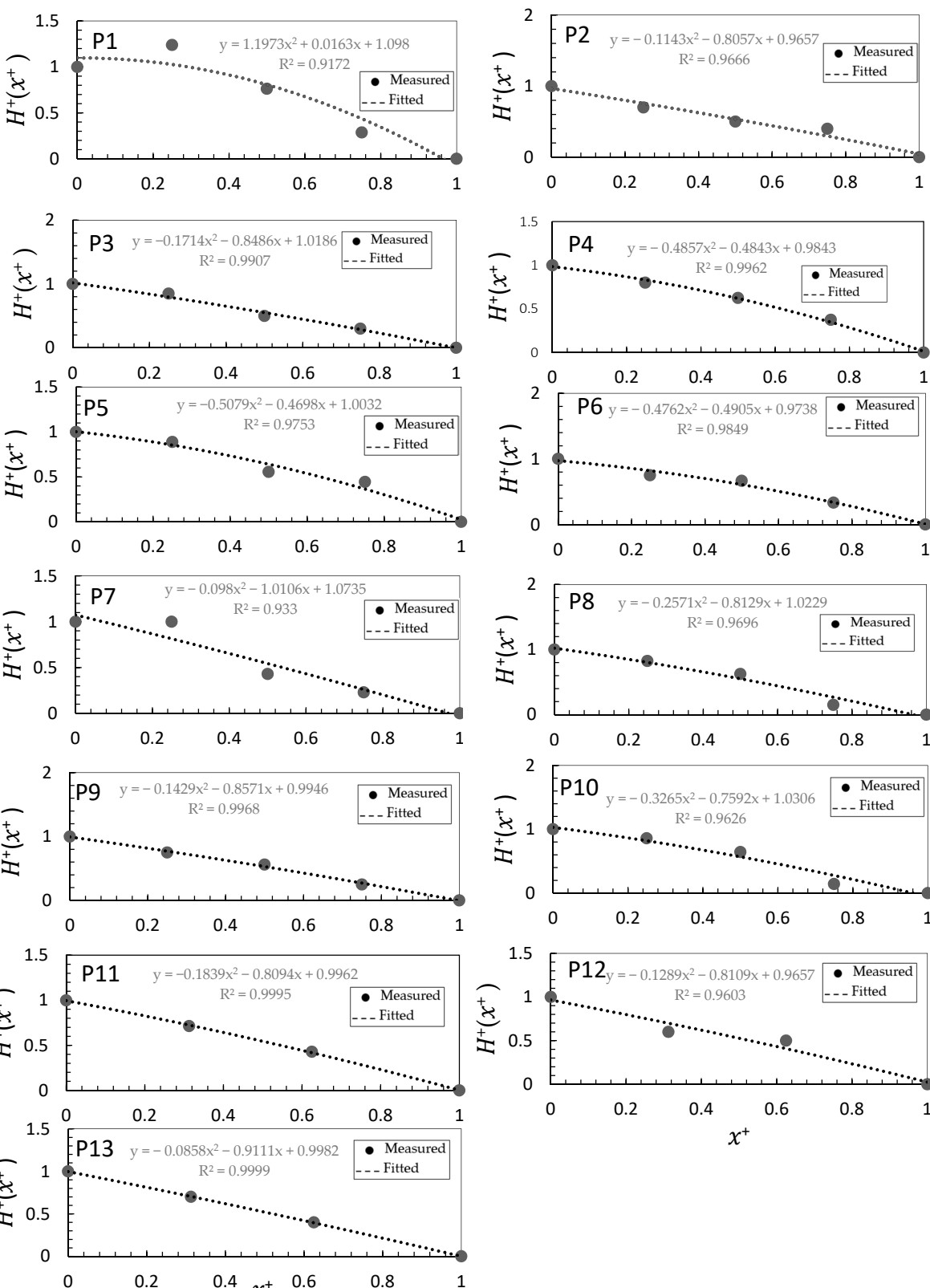

**Figure 4.** Comparison between measured and fitted normalized water surface profiles along the normalized distance (streamwise direction) for all vegetated patches.

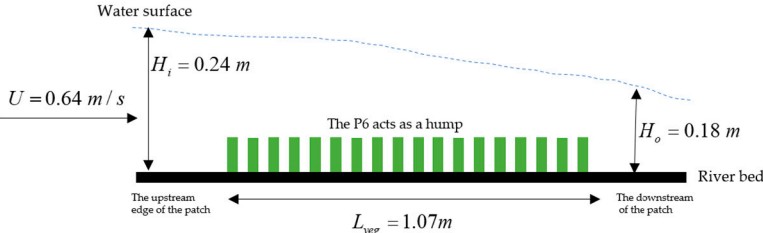

**Figure 5.** The variation in the water surface profile over P6.

The variation in $H(x)$ in the presence of the vegetation patches can be explained as follows: each patch acts as a hump on the channel bed, as shown in Figure 4. As we know, the hydraulics for a rectangular channel can be expressed as [42]

$$\frac{dE}{dx} + \frac{dz}{dx} = 0 \ \ or \ \ \frac{dy}{dx} \times \frac{dE}{dy} + \frac{dz}{dx} = 0 \tag{12}$$

where $y$ is the vertical distance from the channel bed $z$ and $E = y + U^2/2g$ is the specific energy [42]. Considering $dE/dy = 1 - U^2/yg = 1 - Fr^2$, Equation (12) can be written as

$$\left(1 - Fr^2\right)\frac{dy}{dx} + \frac{dz}{dx} = 0 \tag{13}$$

As shown in Figure 4, as the vegetation patch acts as a hump on the channel bed, $dz/dx > 0$ along the flume bed. Therefore, from Equation (13), $(1 - Fr^2)dy/dx < 0$. In addition, as the flow is subcritical ($Fr < 1$), the water depth decreases (i.e., $dy/dx < 0$) along the channel with vegetation patches. Figure 4 shows the variation in water depth in the presence of the patch. It should be noted that, as it was difficult to collect data in natural rivers since last year, the focus of field measurements was on the field measurements along river reaches with vegetation patches. In addition, to make sure our field measurements were reliable, measurements at each point were repeated several times to check and take an average. In addition, the duration for collecting data at each point was 50 s in order to obtain a significant velocity result.

### 4.2. Streamwise Distribuation of $C_{dv}$

In this study, the drag coefficient $C_{dv}$ was calculated by using Equation (8). Figure 6 presents the distribution of the drag coefficient $C_{dv}$ against the streamwise distance $x$ for all vegetated patches. However, the variation in the normalized streamwise drag coefficient $C_{dv}(x^+L_{veg})/< C_{dv} >$ is presented in Figure 7. The streamwise averaged drag coefficient $< C_{dv} > = \int_0^1 C_{dv}(x^+L_{veg})d(x^+)$ with $x^+ = x/L_{veg} \leq 1$ was adopted here for comparison with $C_{dv}$ for different values of $\varnothing_{veg}$ and $L_{veg}$. The variations in the drag coefficient for P3 and P4 have an obvious quadratic form compared to those of other vegetated patches. For these two vegetated patches, the variations in drag coefficient $C_{dv}$ show a quadratic form that first increases and then decreases, which is consistent with the results obtained in previous studies on nonuniform flow in the presence of a rigid vegetated patch [31,38]. The results indicate that, for the vegetation patches having an increasing drag coefficient $C_{dv}$ with $x$ (the distance from the upstream edge of the vegetation patch), the intensity is weaker than those for the vegetation patches of P2 and P9. Moreover, the maximum value of the drag coefficient $C_{dv}$ is observed in the trailing edge of the vegetation patch. In Equation (8), $E_{veg}$ is constant for each vegetation patch, which represents the vegetation properties and its distribution on the bed. Thus, the variation in $C_{dv}$ is determined by the terms of $S_0/U^2$ and $(1/U^2 - 1/gH(x))S_f$ of Equation (8), where $S_f = -\partial H/\partial x = -(2ax + b)$. The water depth $H(x)$ decreases gradually with the distance inside the vegetated patch (Figure 3); thus, $S_0/U^2$ and $1/U^2$ decrease and $1/gH(x)$ increases with $x$, and $(1/U^2 - 1/gH(x))$

decreases in the streamwise direction. Meanwhile, $S_f$ increases with $x$ according to the parameters in Table A1 (i.e., $a< 0$, $S_f >0$). Taken together,

$$C_{dv} = Constant\ value \times \left[\downarrow_- \times (1+\uparrow^+)\right] = Constant\ value \times \left[\downarrow_- \times \uparrow^+\right] \tag{14}$$

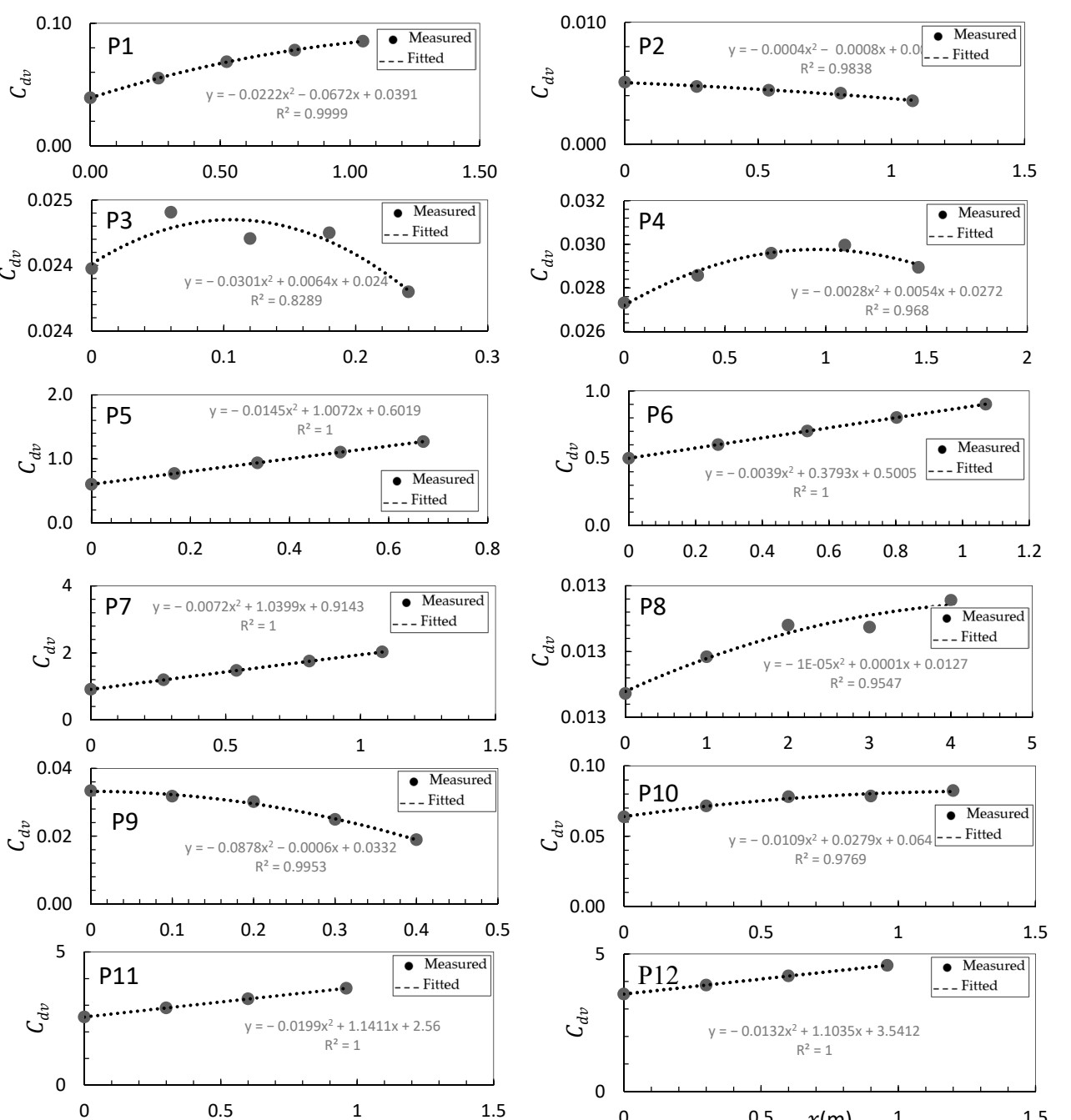

**Figure 6.** The streamwise variations in drag coefficient $C_{dv}$ of the flexible vegetation obtained from the field data for all vegetated patches.

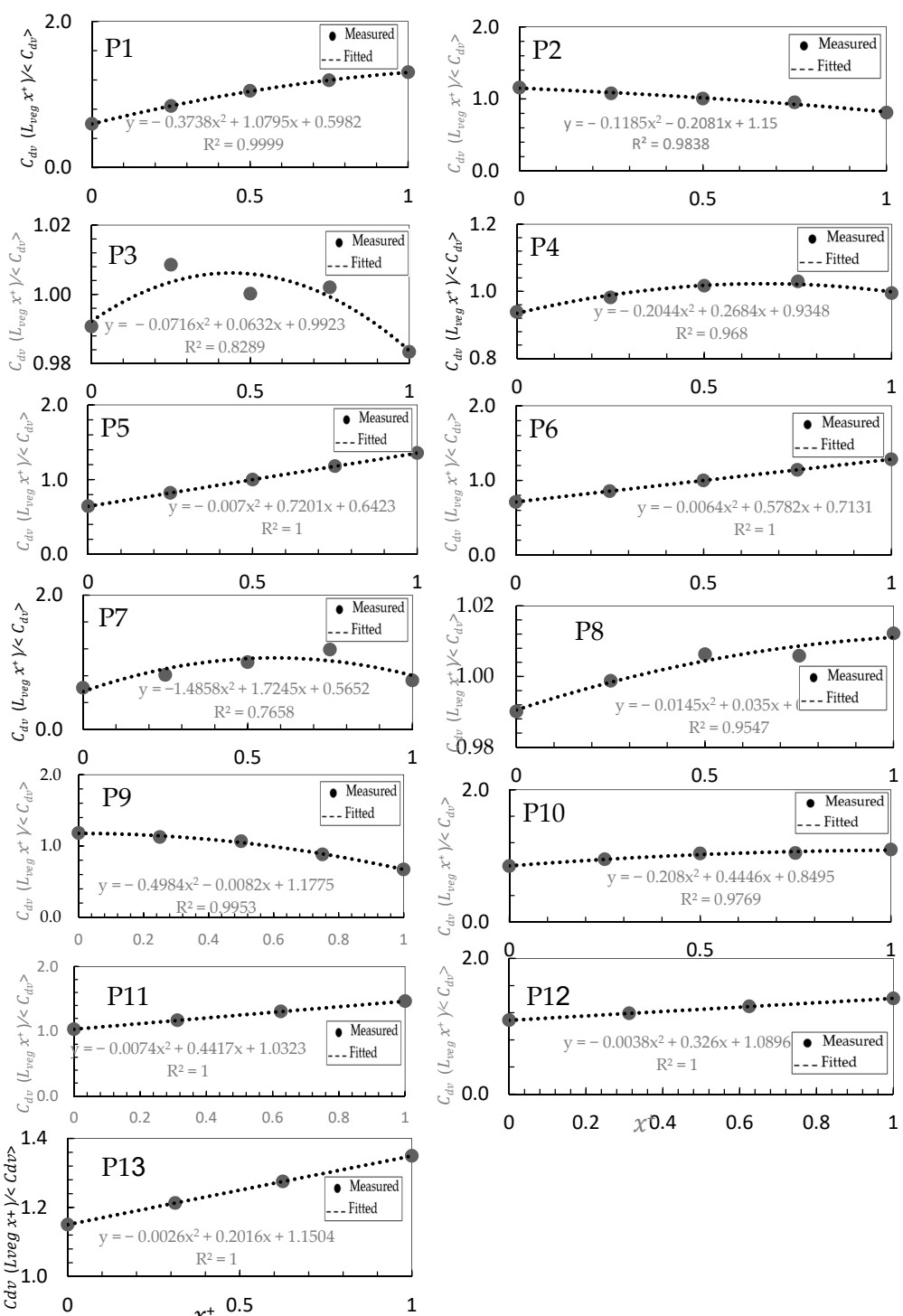

**Figure 7.** Variations in $C_{dv}(x^+ L_{veg})/<C_{dv}>$ with $x^+$.

Equation (14) presents a nonmonotonic result and explains why the drag coefficients $C_{dv}$ and $C_{dv}(x^+ L_{veg})/<C_{dv}>$ of the submerged flexible vegetation in gravel bed rivers show a parabolic form. This phenomenon has also been studied by other researchers [43]. It is obvious that the nonmonotonic change in $C_{dv}$ is due to the effect of nonuniform flow in the river. The Reynolds number of each cylindrical element $Re_d = U(x)D/\vartheta_m$ has

a significant effect on the drag coefficient $C_{dv}$ [43]. The drag coefficient for an isolated cylinder $C_{d-iso}$ can be estimated by following equation [18]:

$$C_{d-iso} = 11Re_d^{-0.75} + 0.9\left[1 - exp\left(-\frac{1000}{Re_d}\right)\right] + 1.2\left[1 - exp\left[-\left(\frac{Re_d}{4500}\right)^{0.7}\right]\right] \quad (15)$$

Figure 8 shows the variation in the drag coefficient $C_{dv}$ with $Re_d$ ranging from 316 to 3757. The drag coefficient $C_{d-iso}$ of isolated vegetation in Figure 8 was estimated by using Equation (16). The drag coefficients $C_{dv}$ for vegetation patches P1 to P4, P6, and P8 to P10 are lower than $C_{d-iso}$, implying a sheltering effect.

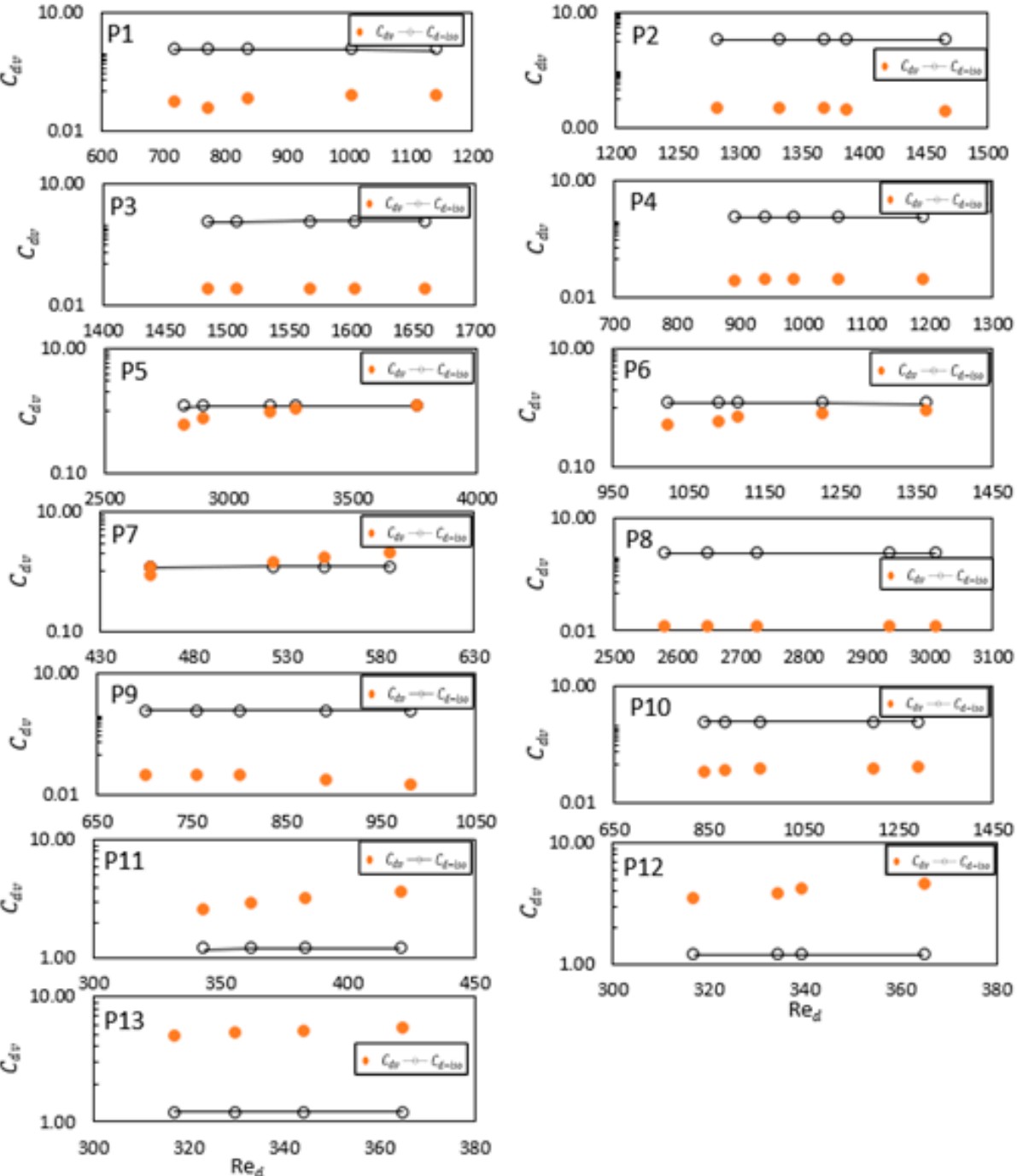

**Figure 8.** Variations in the drag coefficient $C_{dv}$ of flexible vegetation with $Re_d = U(x)D/\vartheta_m$.

On the other hand, for vegetation patches P11 to P13, the drag coefficient $C_{dv}$ is larger than $C_{d-iso}$, indicating a blockage effect. It is observed that, for vegetated patches P5 and P7, the sheltering effect appears near the leading edge, as well as the blockage effect at the tail of these flexible vegetation patches. When the Reynolds number is sufficiently low, the viscous effects cannot be ignored relative to those from drag, and this effect is called the blockage effect [16]. However, for dense flexible vegetation patches, due to the bending deformation of the flexible vegetation, the downstream body is more fully located in the wake region of the upstream body; meanwhile, the bending deformation results in a significant reduction in the spacing distance between the bodies, causing an intensified sheltering effect and a lower form drag force [15]. Furthermore, turbulent fluctuations in the wakes of upstream curved vegetation are stronger than those in the wakes of rigid vegetation, introducing additional kinetic energy into the boundary layer of adjacent downstream vegetation that can delay the flow separation [2]. The staggered arrangement of vegetal elements reinforces this effect [44]. This delayed flow separation tends to reduce the drag coefficient $C_{dv}$ of a vegetation patch relative to that of a single cylindrical element [15]. When $W_{veg}$ is low (P11 to P13), the flow nonuniformity is not that obvious, and the bending deformation of the vegetation patch is reduced. Thus, the sheltering effect becomes weaker, and the blockage effect becomes stronger.

*4.3. Drag Coefficient Factor $\alpha_b$*

In the following section, in order to investigate the effect of reconfiguration and deformation on the drag of the flexible vegetation patches, the drag coefficient factor $\alpha_b = C_{dv}/C_d$ was employed [45]:

$$C_d = 0.819 + \frac{58.5}{\sqrt{Re_v}} \tag{16}$$

where $Re_v$ is the vegetation Reynolds number about a vegetation-related hydraulic radius $R_v$ and is defined as [18]

$$Re_v = \frac{U(x)R_v}{\vartheta_m} = \frac{\pi}{4} \times \frac{U(x)}{\vartheta_m} \times \frac{D(1 - \alpha\varnothing_{veg})}{\alpha\varnothing_{veg}} \tag{17}$$

where $\alpha = \overline{h}_p/H(x)$. Figure 9 shows a comparison between $C_{dv}$ and $C_d$ for each vegetation patch with a specified density of $\varnothing_{veg}$ with a Reynolds number of $Re_v$ in the range of $0.03 \times 10^6 < Re_v < 61 \times 10^6$.

Figure 10 presents the variations in the drag coefficient factor $\alpha_b = C_{dv}/C_d$ with $Re_v$, ($\alpha_b > 1$ shows a blockage effect, while $\alpha_b < 1$ shows a sheltering effect). For most of the vegetation patches in this study, $\alpha_b < 1$, implying the sheltering effect is predominant. However, for vegetation patches P11, P12, and P13, the blockage effect is observed. Generally, the greater the bending deformation, the stronger the sheltering effect, and the more delayed the flow separation in the boundary layer.

Both bending and bifurcation are observed at the upstream and downstream edges of each vegetation patch in the field. These two features have a significant effect on the drag reconfiguration. Figure 11 shows the variations in the drag coefficient factor $\alpha_b$ with $Re_d$. For vegetation patches P3, P4 and P8, the value of $\alpha_b$ gradually increases from the leading edge of the vegetation patch and decreases after reaching the peak value until arriving at the tail of the vegetation patch. For patches P2, P9, and P11, the maximum value of $\alpha_b$ gradually decreases as one approaches the downstream edge of the vegetation patch; and for remaining vegetation patches, $\alpha_b$ gradually increases from the leading edge of the vegetation patch and reaches the peak value at the downstream edge of the vegetation patch.

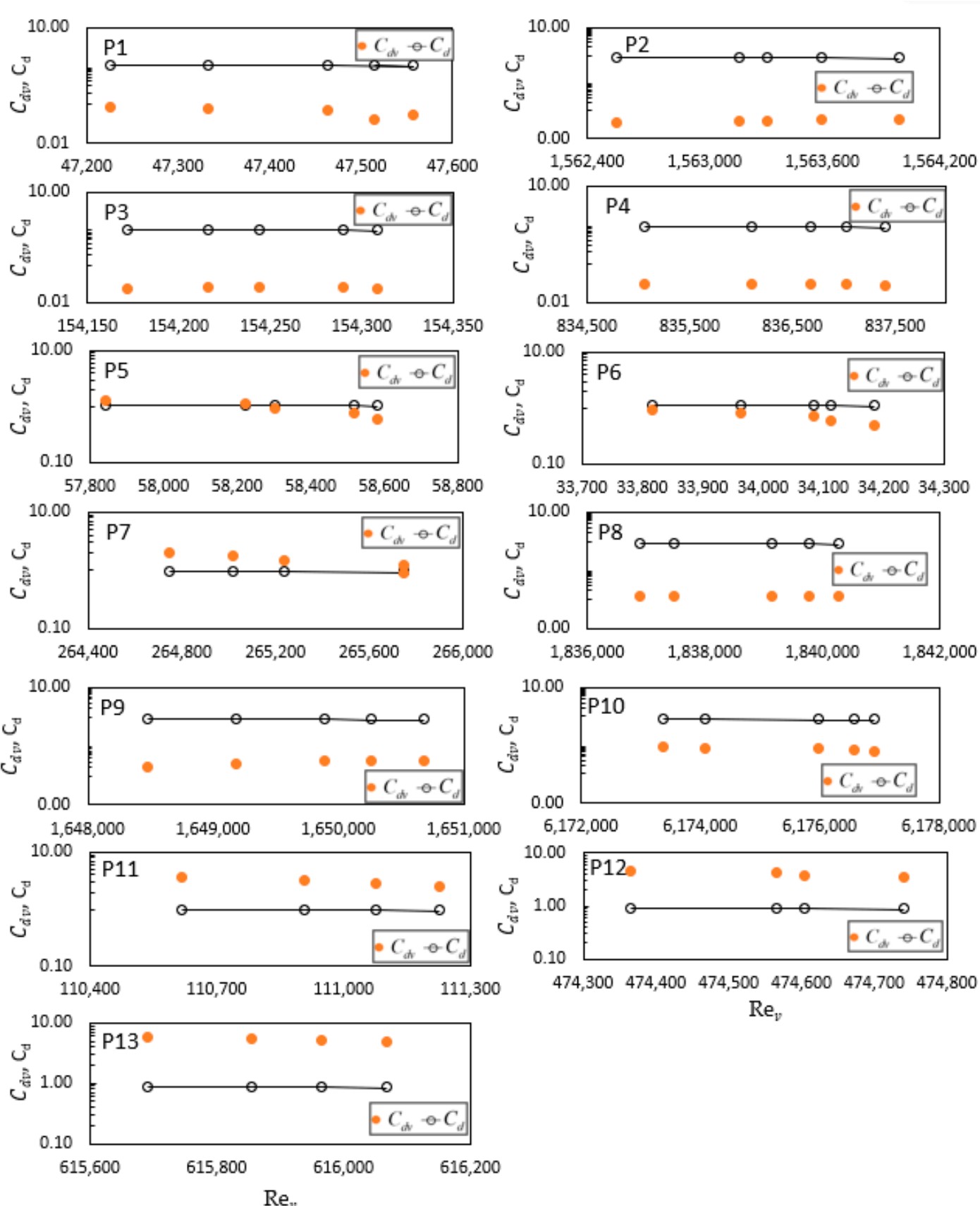

**Figure 9.** Variations in $C_{dv}$ and $C_d$ with $Re_v$.

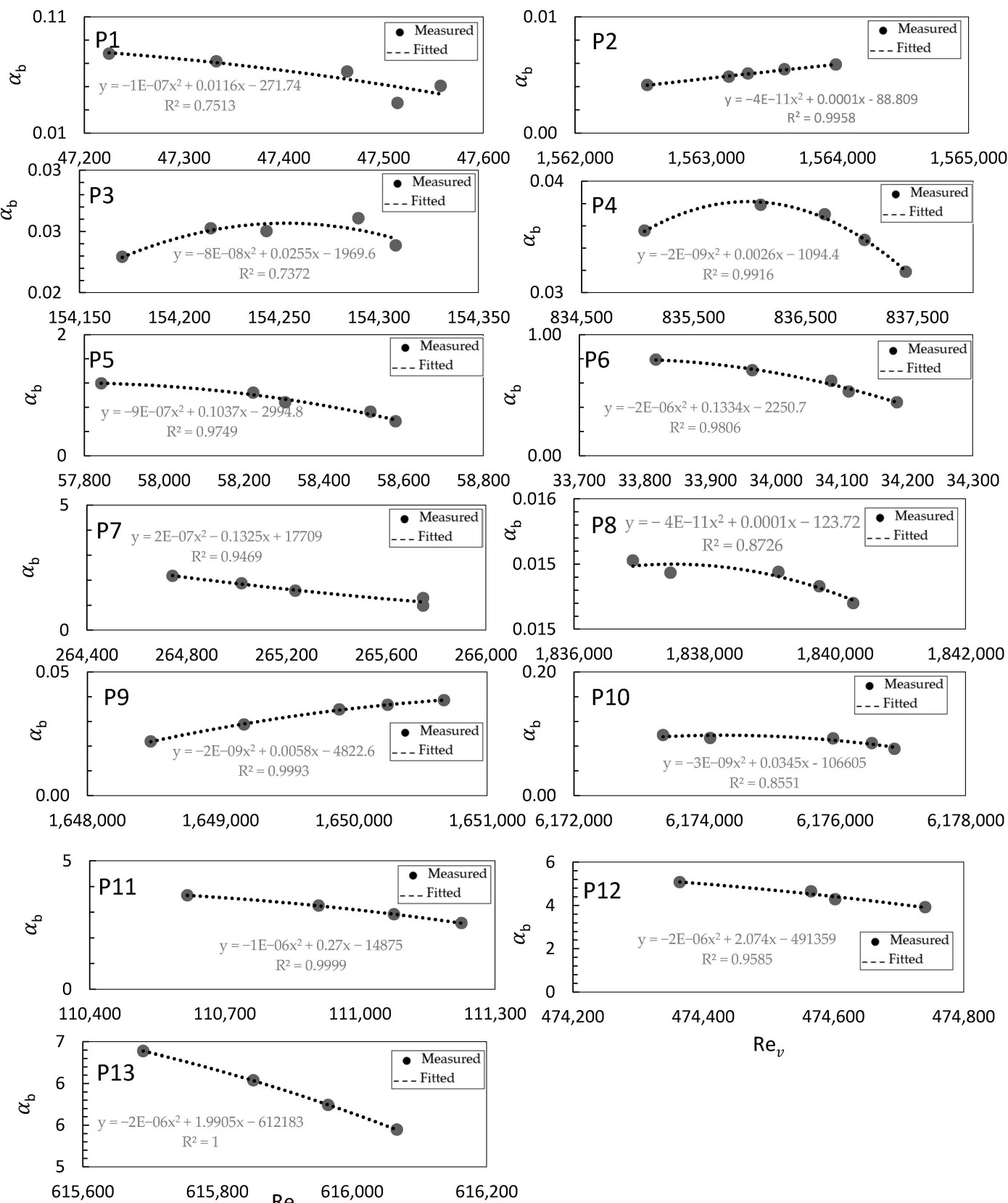

**Figure 10.** Variations in the drag coefficient factor $\alpha_b$ with $Re_v$.

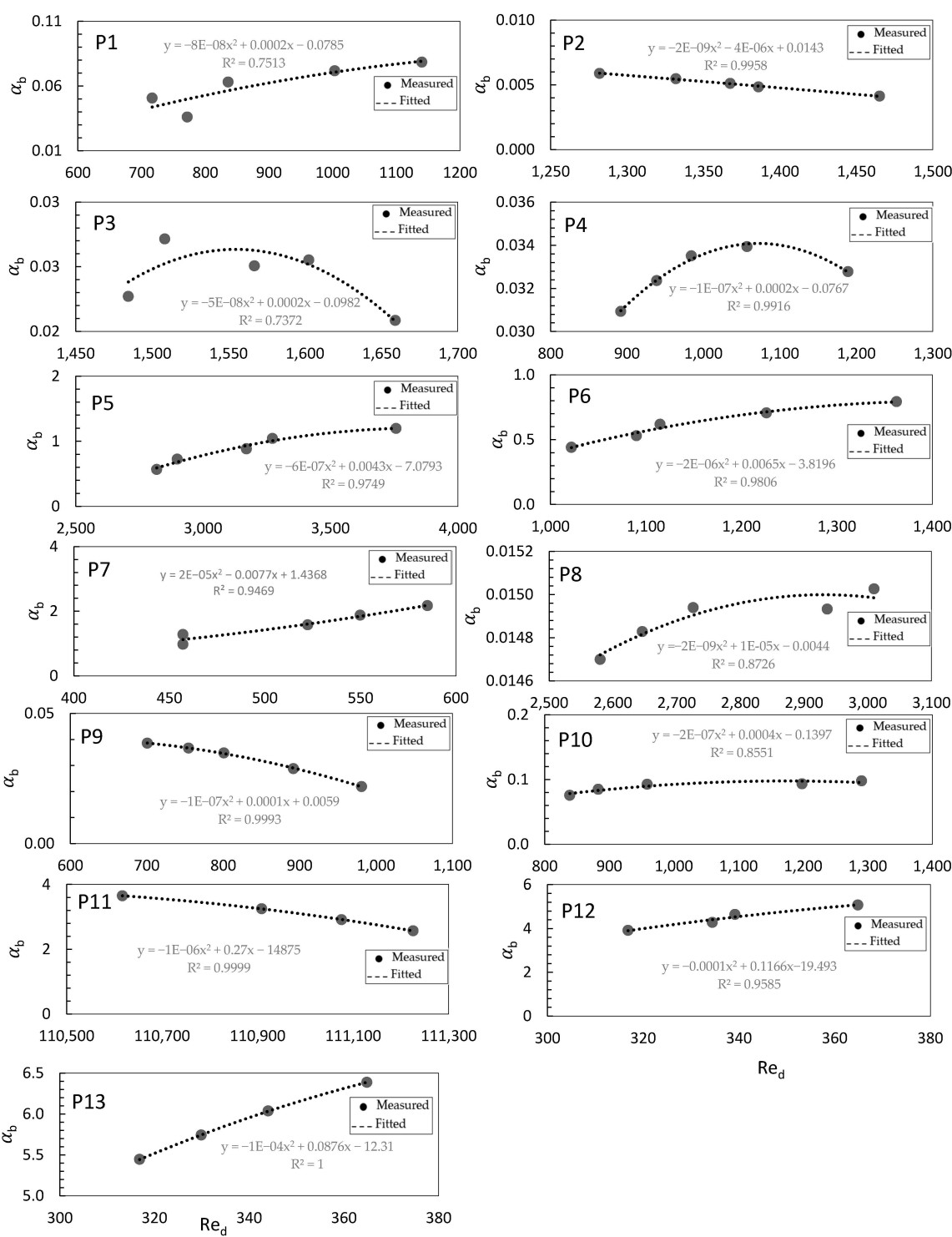

**Figure 11.** Variations in the drag coefficient factor $\alpha_b$ with $Re_d$.

### 4.4. Analysis of $\alpha_b$ Using the Cauchy Number $C_s$

The forces exerted on a plant change as its bending continues [37]. In this section, the relationship between $\alpha_b$, which is used to describe the bending of flexible elements, and the mechanical properties of plant elements ($C_s$) is investigated. The Cauchy number was determined based on Equation (8). Figure 12 shows the nonmonotonic tendency of change in $\alpha_b$ with respect to $C_s$ for all vegetation patches. For all vegetation patches (except P2 and P9), $\alpha_b$ increases with $C_s$ and, after reaching the maximum value, then decreases (specially for P3 and P4), which is consistent with the results of Zhang et al. (2020) for emergent

flexible vegetated patches [38]. The relationship between $\alpha_b$ and $C_s$ for submerged flexible vegetation patches in gravel bed rivers can be described as follows:

$$\alpha_b = -1.023\text{Ln}(C_s) - 5.4 = -1.023\text{Ln}\left(\frac{\rho U^2}{E_s}\right) - 5.4 \quad R^2 = 0.8849 \tag{18}$$

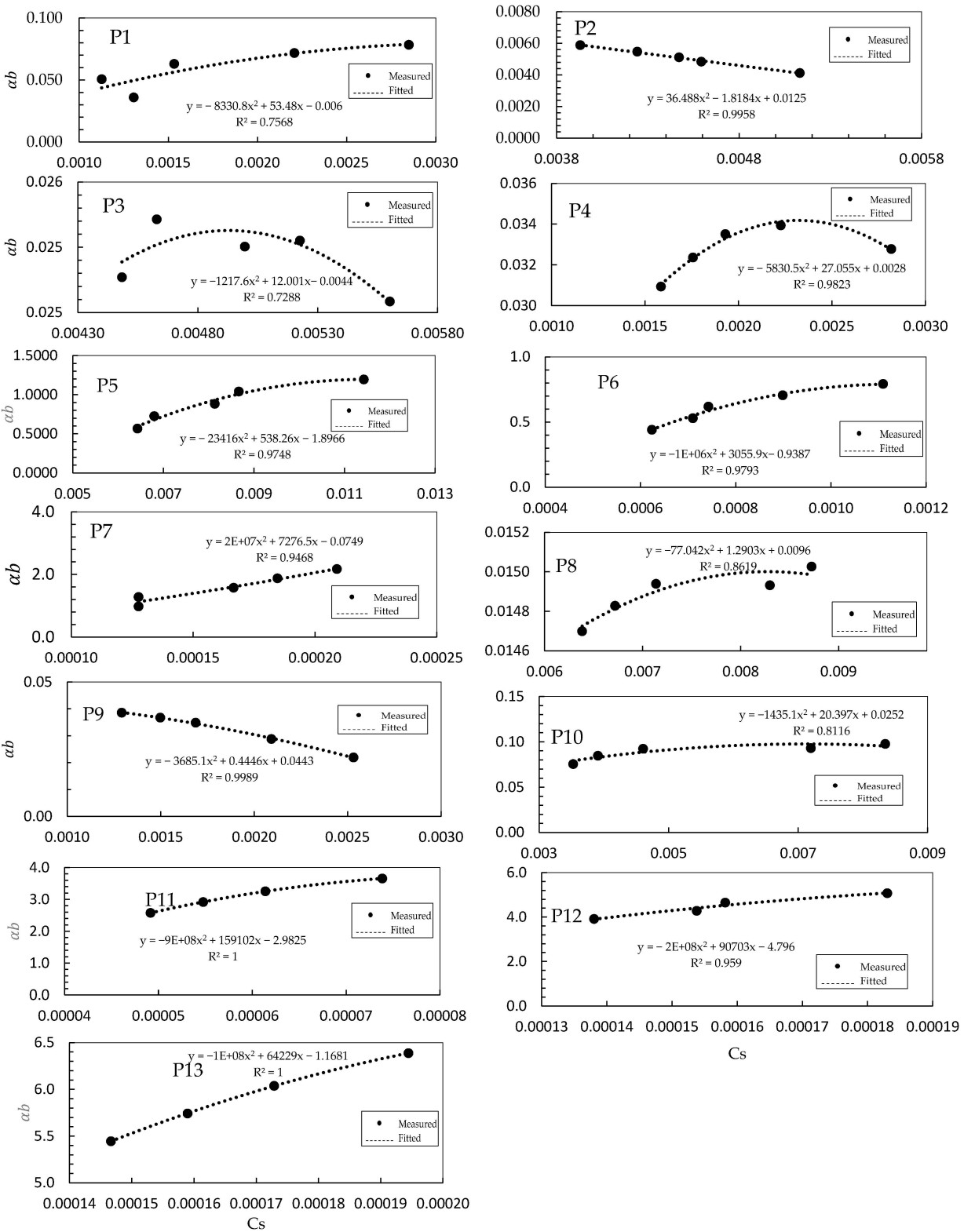

**Figure 12.** Variations in the drag coefficient factor $\alpha_b$ with $C_s$.

Up until now, only a few experimental studies have been conducted in the laboratory by using emergent elements [38,45]. Equation (18) is developed based on the velocity $U$ and Young's modulus $E_s$ in rivers where many uncontrolled factors may affect $\alpha_b$ and the value of determination coefficient ($R^2$). The estimation of the $E_s$ value for flexible submerged vegetation or other elements remains a challenge in field studies without advanced instruments. The $E_s$ value can be estimated by assuming a homogeneous element and measuring its deflection under a load [39]. However, in this study, it is estimated by Equation (11), as shown in Section 2.3 [46]. Equation (18) presents the new fitted formula for submerged flexible vegetation patches in gravel bed rivers. A Cauchy number $C_s$ ranging from $1.49 \times 10^{-4}$ to $74.4 \times 10^{-4}$ was used to incorporate the effects of flexibility into the flexible drag force based on the drag coefficient factor.

## 5. Discussion

Up until now, most studies on the drag coefficient of vegetation patches have been limited to laboratory experiments with either rigid or flexible elements [36,38,45]. Research work on the drag coefficient in the presence of submerged flexible vegetation in gravel bed rivers is very important. One can see from Figure 5 that for nonuniform flows with a large hydraulic gradient, the predicted water surface profiles using Equation (9) agree well with those measured in field. However, for nonuniform flows in gravel bed rivers with a low hydraulic gradient, the regression method has certain limitations due to the variation in roughness from gravel to vegetation. This variation is considered by estimation of $\partial U / \partial x$ [36,47].

The variation in the normalized streamwise drag coefficient $C_{dv}(x^+ L_{veg}) / < C_{dv} >$ is presented in Figure 7. The streamwise averaged drag coefficient $< C_{dv} > = \int_0^1 C_{dv}(x^+ L_{veg}) d(x^+)$ with $x^+ = x / L_{veg} \leq 1$ is adopted here for comparison with $C_{dv}$ for different values of $\varnothing_{veg}$ and $L_{veg}$. It is found that two different patterns can be distinguished based on the area intensity of vegetation ($\varnothing_{veg}$). For vegetation patches P3 and P4, $C_{dv}(x^+ L_{veg}) / < C_{dv} >$ increases first and then decreases with the change in $x^+ = x / L_{veg}$, forming an obvious crescent shape. For other vegetation patches, although the variation in the streamwise-averaged drag coefficient is a nonmonotonic form, the level of change is much weaker than those for P3 and P4.

The results of previous studies have indicated that, for the rigid emergent vegetation patches in a uniform flow, the blockage effect occurs as $Re_d < 7000$ [15] and the blockage effect in a nonuniform flow is weaker than that in a uniform flow [31]. However, the flexible vegetation patches in the nonuniform flow in gravel bed rivers are completely different from the rigid vegetation patches used in laboratory experiments. In the present study, the blockage effect appears as $Re_d < 400$, and the sheltering effect is observed for $Re_d > 580$. Both the blockage effect and sheltering effect appear in the range of $450 < Re_d < 500$, especially for vegetation patches P5 and P7. This is shown in Figure 9; for vegetation patches P1 to P4, P6, and P8 to P10, $C_{dv} < C_d$; and for P11 to P13, $C_{dv} > C_d$. Meanwhile, for patch P5 and at the upstream edge of the patch to the middle part of the patch, $C_{dv} > C_d$, and then from this point to the downstream edge of the patch, $C_{dv} < C_d$. This result is partly due to the reduction in drag caused by the bending effect of flexible vegetation as most elements of the vegetation patch tend to be bifurcate.

In Figure 9, for all vegetation patches, the change in $C_{dv}$ with $Re_v$ has different trends, which is different from the monotonic trend of $C_d$, confirming the results obtained by Zhang et al. (2020) on the emergent flexible vegetation patches [38]. Two effects caused by flexible vegetation patches are observed in Figure 11, namely, the sheltering effect and blockage effect, which occur for $Re_d > 580$ and for $Re_d < 450$, respectively. In most of the vegetated patches, the sheltering effect is dominant, leading to a decrease in the drag coefficient. Finally, a fitting formula is proposed for determining the drag coefficient and Cauchy number in Equation (18). The $C_s$ values indicating the magnitude of the reconfiguration of a flexible vegetation patch depend on the flow conditions and plant characteristics. The factor of drag coefficient $\alpha_b$ explains how drag is affected by forces acting on flexible, deformed, or reorganized vegetation for the same projected area [45]. It is noted that the

reduction in drag for flexible vegetations is represented by $\alpha_b$, which is the integrated adjustment factor including the distortion of the vegetation area due to its bending [38].

## 6. Conclusions

In gravel-bed rivers, flexible vegetation patches are frequently observed, showing the key role that vegetation patches play in the river dynamics and sediment transportation. This calls for more research in rivers on submerged flexible vegetation patches rather than under controlled conditions in a laboratory. To our knowledge, the reported studies are either limited to laboratory experiments or emergent vegetal elements. The present study was conducted to investigate the variations in the drag coefficient of submerged flexible vegetation patches in gravel rivers. The Saint-Venant equation was applied to estimate the drag coefficient $C_{dv}$. As the water surface profile is one of the most important parameters in calculating $C_{dv}$, the water surface profiles along several reaches of the Padena Marbor and Beheshtabad gravel bed rivers were measured along each vegetation patch. The following conclusions can be drawn from this study:

1. The nonmonotonic change in $C_{dv}$ is due to the effect of flow nonuniformity in the rivers. In addition, the nonuniformity of the grain size distribution around each vegetated patch influences this nonmonotonic variation in $C_{dv}$.
2. The maximum value of the drag coefficient $C_{dv}$ is observed near the trailing edge of the vegetation patch in gravel-bed streams.
3. The drag coefficient $C_{dv}$ for vegetation patches, e.g., P1 to P4, P6, and P8 to P10, is lower than the drag coefficient of isolated vegetation $C_{d-iso}$, implying a sheltering effect. On the other hand, for some vegetation patches, e.g., P11 to P13, the drag coefficient $C_{dv}$ is larger than $C_{d-iso}$, indicating a blockage effect.
4. The blockage effect appears as $Re_d < 400$, and the sheltering effect is observed for $Re_d > 580$. Both the blockage effect and sheltering effect appear in the range of $450 < Re_d < 500$, especially for vegetation patches P5 and P7. Meanwhile, the bending deformation results in a significant reduction in the spacing distance between the bodies, causing an intensified sheltering effect and a lower form drag force.
5. The drag coefficient factor $\alpha_b$ changes with the Cauchy number $C_s$ in the streamwise flow direction. The variation in this factor is not stable from the leading to the trailing edge, showing a very complex flow pattern along a vegetation patch in rivers.
6. Equation (18) can be used to predict the drag coefficient factor for submerged flexible vegetation patches in gravel bed rivers by using the velocity $U$ and Young's modulus $E_s$. However, more data considering the boundary layer concept are needed along submerged flexible vegetation patches in rivers to predict a better drag coefficient.

The results of this study will help the designers and engineers to take into account the effect of submerged flexible vegetation patches and flow nonuniformity in drag coefficient estimation in natural streams.

**Author Contributions:** K.N., field works, methodology, software, writing—original draft, preparation; H.A., supervision, methodology, validation; J.S., methodology, writing—review and editing. All authors have read and agreed to the published version of the manuscript.

**Funding:** This research received no external funding.

**Institutional Review Board Statement:** Not applicable.

**Informed Consent Statement:** Not applicable.

**Data Availability Statement:** Data are contained within the article and Appendix A.

**Acknowledgments:** The authors thank Marwan Hassan, a professor at the University of British Columbia, Canada, for language edits, and Ruhollah Nouri pour, a water engineering graduate from Shiraz University in Iran, for topographic survey in Rivers.

**Conflicts of Interest:** The authors declare no conflict of interest.

## Appendix A

**Table A1.** Parameters of each vegetal patch.

| Patch | $\varnothing_{veg}$ | $E_{veg}$ (m) | $H_i$ (m) | $H_1$ (m) | $H_2$ (m) | $H_3$ (m) | $H_4$ (m) | $H_5$ (m) | $H_o$ (m) | a | b | c | $R^2$ | $B$ (m) | $U$ (m/s) | $Q$ (m³/s) | $S_0$ (m) |
|---|---|---|---|---|---|---|---|---|---|---|---|---|---|---|---|---|---|
| P1 | 0.0341 | 0.44 | 0.13 | 0.14 | 0.12 | - | - | 0.1 | 0.088 | −0.0456 | 0.0007 | 0.1341 | 0.9172 | 3.63 | 0.67 | 0.32 | 0.0402 |
| P2 | 0.0186 | 0.65 | 0.16 | 0.154 | 0.15 | - | - | 0.148 | 0.14 | −0.002 | −0.0149 | 0.1593 | 0.9666 | 3.63 | 1.44 | 0.84 | 0.0309 |
| P3 | 0.0303 | 0.54 | 0.19 | 0.187 | 0.18 | - | - | 0.176 | 0.17 | −0.0595 | −0.0707 | 0.1904 | 0.9907 | 3.63 | 1.167 | 0.80 | 0.042 |
| P4 | 0.0201 | 0.60 | 0.16 | 0.152 | 0.145 | - | - | 0.135 | 0.12 | −0.0091 | −0.0133 | 0.1594 | 0.9962 | 3.63 | 0.982 | 0.57 | 0.04 |
| P5 | 0.0655 | 0.29 | 0.18 | 0.175 | 0.16 | - | - | 0.155 | 0.135 | −0.0509 | −0.0316 | 0.1801 | 0.9753 | 3.85 | 1.01 | 1.25 | 0.029 |
| P6 | 0.0469 | 0.34 | 0.24 | 0.225 | 0.22 | - | - | 0.2 | 0.18 | −0.025 | −0.0275 | 0.2384 | 0.9849 | 4.8 | 0.64 | 0.93 | 0.038 |
| P7 | 0.0203 | 0.61 | 0.32 | 0.32 | 0.28 | - | - | 0.266 | 0.25 | −0.0059 | −0.0655 | 0.3251 | 0.933 | 4.5 | 0.6 | 0.7 | 0.037 |
| P8 | 0.0254 | 0.53 | 0.28 | 0.273 | 0.265 | - | - | 0.246 | 0.24 | −0.0006 | −0.0081 | 0.2809 | 0.8452 | 4.42 | 1.091 | 3.09 | 0.021 |
| P9 | 0.0186 | 0.65 | 0.28 | 0.26 | 0.245 | - | - | 0.22 | 0.2 | −0.0714 | −0.1714 | 0.2796 | 0.9968 | 3.88 | 1.365 | 0.847 | 0.041 |
| P10 | 0.0142 | 0.75 | 0.4 | 0.38 | 0.35 | - | - | 0.28 | 0.26 | −0.0317 | −0.0886 | 0.4043 | 0.9626 | 5.09 | 1.07 | 2.179 | 0.04 |
| P11 | 0.0321 | 0.42 | 0.38 | 0.35 | 0.34 | 0.36 | 0.35 | 0.34 | 0.31 | −0.1839 | −0.8094 | 0.9962 | 0.995 | 8.16 | 0.36 | 0.99 | 0.008 |
| P12 | 0.0233 | 0.58 | 0.38 | 0.35 | 0.36 | 0.34 | 0.33 | 0.33 | 0.33 | −0.1289 | −0.8109 | 0.9657 | 0.9603 | 8.16 | 0.36 | 0.99 | 0.008 |
| P13 | 0.0191 | 0.71 | 0.38 | 0.34 | 0.35 | 0.365 | 0.33 | 0.33 | 0.33 | −0.0858 | −0.9111 | 0.9982 | 0.9999 | 8.16 | 0.36 | 0.99 | 0.008 |

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
