# Peer review of "Drag Coefficient of Submerged Flexible Vegetation Patches in Gravel Bed Rivers"

_water, doi:10.3390/w14050743_

Round 1

Reviewer 1 Report

WATER

Drag Coefficient of Submerged Flexible Vegetation Patches in Gravel Bed Rivers

By Kourosh Nosrati, Hossein Afzalimehr and Jueyi Sui

The paper illustrates the results of a field campaign carried out to evaluate the drag coefficient of submerged flexible vegetation in gravel bed rivers.

Although the topic is of some interest to the scientific community, the paper needs some improvements.

Main issues:

  1. In my opinion the highlight of the paper is the field campaign. Therefore, a section which describes the sites, the characteristics of the patches, the experimental setups and the duration of the data acquisition is requested.
  2. Authors should also demonstrate the reliability of their measurements. for example, a comparison with data collected outside the patches is required.
  3. The introduction should also describe the results of other relevant paper focused on the influence of blade flexibility on the drag coefficient both in current and wave conditions. Hereafter a paper that may be worth quoting are reported:
  • Houser, C., Trimble, S., & Morales, B. (2015). Influence of blade flexibility on the drag coefficient of aquatic vegetation. Estuaries and Coasts, 38(2), 569-577. doi:10.1007/s12237-014-9840-3
  • Luhar, M., Infantes, E., & Nepf, H. (2017). Seagrass blade motion under waves and its impact on wave decay.Journal of Geophysical Research: Oceans, 122(5), 3736-3752. doi:10.1002/2017JC012731
  • Cavallaro, L., Viviano, A., Paratore, G., & Foti, E. (2018). Experiments on surface waves interacting with flexible aquatic vegetation.Ocean Science Journal, 53(3), 461-474. doi:10.1007/s12601-018-0037-8

 Minor points:

  1. Eq (3) - Please define H
  2. Eq (4) - Please define Cdv
  3. Eq (5) - Please define B
  4. Table 1 - Please check the unit of measure of U
  5. Table 1 - Please give the information about “B” instead of “B/H”
  6. Table 1 - Please explain how did you measure or estimate the parameters
  7. 215 – add “m” after “downstrea”
  8. 252 – add”n” before “onuniform”
  9. 366 – change “thst” with “that”

Author Response

Please see attached discussion letter.

Reviewer 2 Report

Dear Authors,

The research you have presented, analysis of the drag coefficient for submerged flexible vegetation patches based on the field measurements is very interesting. Unfortunately, the quality of the manuscript doesn’t match the quality of the research topic. Therefore, the manuscript needs to be significantly improved before publishing. The main issue of the paper is inappropriate self-referencing by the authors and lack of the background for the research and conclusions.

Please find below main shortcomings that need to be addressed:

Out of 51 references used in the paper, 13 are from the authors themselves, which is more than 25%. Additionally, authors haven’t referenced their work appropriately - in the state of the art review – they used references for the general statements, e.g. “plants play important role in transporting contaminants…” (3 references), “vegetation patches have numerous benefits for the environment” (4 references), etc.

State-of-the art review is non-existent. There is no review of the current research that would provide background for the field and context for the presented research. The majority of the references are grouped together in dummy-referencing (16 references in the 3 sentences of the Introduction section), without explaining the individual contribution for each of them.

There are several equations that are presented, but not followed through:

  • Saint-Venant equation presented in the paper (eq. 3) is deemed as “S-V equation for an open channel flow through vegetation patch”, and referenced from recent publications, but it is Saint-Venant equation of as standard form. Why is it presented as a novel, vegetation-related equation?
  • In a equation (7) derivative of H is presented, which is not used later on in the paper. What is it’s value?

Several statements are not quantified:

  • Application of the quadratic regression formula that is used in the paper is explained as “not appropriate for low hydraulic gradients”, but there is no quantification of the cut-off boundary for low gradient.
  • Please reference the accuracy of the “Butterfly Current Meter” from the manufacturer’s user manual.
  • “The flow rate is variable for each vegetation patch” – explain how is flow rate associated with specific flow patch from the total Q in the river?

Paper is missing clear methodological approach to the research - it is rather set of information scattered across the manuscript:

  • Table 2 presents the parches across the 4 study areas from two rivers. However, the association between the Patches and rivers is not explained and it is impossible to understand the context and similarities between the case studies.
  • Why is the water surface slope modelled by the quadratic function? Please explain/reference how was this determined to be appropriate
  • Figure 4 and Figure 5 present comparable data about the drag coefficient, but Figure 4 is depicted with actual lengths, while Figure 5 is depicted with normalized lengths, making the comparison impossible.
  • The relationship between ab and Cs (equation 14) is supposedly done for the entire set of data (?) but the analysis itself is not presented to the readers
  • Overall, results are discussed separately for each patch, without combined analysis and conclusions spanning beyond the case study.
  • Manuscript does not contain any data to support conclusions #1 and #6

There are several typos:

  • uesed
  • Usuage
  • Cauch
  • Padena Marbor =? Padna Marbar
  • Table 3 is referenced on two occasions, probably table 3A?
  • Polynomial fitting in the legend is captioned “measured”, which is not correct. While it presents function of the measured data, it is fitted, not measured.

Author Response

Please see attached discussion letter

Reviewer 3 Report

Dear Authors,

your work is interesting and could help in dealing with submerged flexible vegetation. However, in my opinion, the present version is not yet ready for publication.

Below you can find some very general comments, while I have attached a commented pdf with more specific comments. I hope that my review will help you in better structuring the manuscript.

Introduction

I suggest expanding the literature review on the presence of flexible vegetation in rivers having a mobile bed. This could help in better pinpointing the study novelty.

Material and Methods

I suggest adding more details on how the data were measured, as their quality (and therefore the overall quality of the study) derives also from the used devices. Adding a map of the study area could also help in better understanding where the study was performed.

Results and Discussion

As a general rule, I prefer to see two separated sections (Results + Discussion), using the first to just present the results, and the latter to compare the study with literature evidence.

In Figure 3, are all the data interpolated with the same equation? If yes, I suggest updating the legend naming this equation (its number in the text). A similar hint applies to the other figures.

Author Response

Please see attached discussion letter

Round 2

Reviewer 1 Report

WATER

Drag Coefficient of Submerged Flexible Vegetation Patches in Gravel Bed Rivers

By Kourosh Nosrati, Hossein Afzalimehr and Jueyi Sui

The paper illustrates the results of a field campaign carried out to evaluate the drag coefficient of submerged flexible vegetation in gravel bed rivers.

Although the authors improve the quality of the paper, they did not adequately reply to some of my previous issues.

Main issues:

  1. Authors should also demonstrate the reliability of their measurements. For example, a comparison with data collected outside the patches is required. All the analyses presented derive from the assessment of the depth of the water. Therefore, the authors should show that in their experiments the variation in water depth is due to the presence of the patches.
  2. The introduction should also describe the results of other relevant paper focused on the influence of blade flexibility on the drag coefficient both in current and wave conditions. Hereafter a paper that may be worth quoting are reported:
  • Houser, C., Trimble, S., & Morales, B. (2015). Influence of blade flexibility on the drag coefficient of aquatic vegetation. Estuaries and Coasts, 38(2), 569-577. doi:10.1007/s12237-014-9840-3
  • Luhar, M., Infantes, E., & Nepf, H. (2017). Seagrass blade motion under waves and its impact on wave decay.Journal of Geophysical Research: Oceans, 122(5), 3736-3752. doi:10.1002/2017JC012731
  • Cavallaro, L., Viviano, A., Paratore, G., & Foti, E. (2018). Experiments on surface waves interacting with flexible aquatic vegetation.Ocean Science Journal, 53(3), 461-474. doi:10.1007/s12601-018-0037-8

  1. If the accuracy of the throttle current meter, it makes no sense to show the two decimal digits for the speed data in Table 2.
  2. L 307 and figure 3 – Please use a different symbol to describe the normalized flow depth.

Minor points:

  1. L 103 – replace Ubχ with “Ub
  2. L 110 – Please define “Uχ"
  3. 239 - Please change “vegeation” with “vegetation”
  4. 254 – add “m” after “downstrea”

Author Response

Please see the discussion letter

Reviewer 2 Report

Dear authors,

Thank you for significantly updating your manuscript.

However, my main concerns regarding inappropriate self-referencing and lack of general conclusions linking separate case studies remain unaddressed in this revised manuscript. I suggest that you address this issues before publication in order to increase the quality of the paper. 

Author Response

Please see the discussion letter

Reviewer 3 Report

Dear Authors,

thank you very much for having addressed my comments.

Some minor changes are still needed: I suggest adding the equation number in the legend of Figs. 4,5,8,9,10 as made for Fig. 3

Author Response

Please see the discussion letter

Round 3

Reviewer 1 Report

WATER

Drag Coefficient of Submerged Flexible Vegetation Patches in Gravel Bed Rivers

By Kourosh Nosrati, Hossein Afzalimehr and Jueyi Sui

The paper illustrates the results of a field campaign carried out to evaluate the drag coefficient of submerged flexible vegetation in gravel bed rivers.

The authors addressed my previous issues.